# Parsimonious Effect of Pentoxifylline on Angiogenesis: A Novel Pentoxifylline-Biased Adenosine G Protein-Coupled Receptor Signaling Platform

**DOI:** 10.3390/cells12081199

**Published:** 2023-04-20

**Authors:** William Khoury, Ryan Trus, Xingyu Chen, Leili Baghaie, Mira Clark, Myron R. Szewczuk, Mohammad El-Diasty

**Affiliations:** 1School of Medicine, Queen’s University, Kingston, ON K7L 3L4, Canada; wkhoury@qmed.ca (W.K.); amy.chen@queensu.ca (X.C.); 2Faculty of Arts and Science, Queen’s University, Kingston, ON K7L 3N9, Canada; ryan.trus@med.uvm.edu (R.T.);; 3School of Medicine, The Larner College of Medicine, University of Vermont, Burlington, VT 05405, USA; 4Department of Biomedical & Molecular Sciences, Queen’s University, Kingston, ON K7L 3N6, Canada; 16lbn1@queensu.ca; 5Division of Cardiac Surgery, Queen’s University, Kingston, ON K7L 2V7, Canada

**Keywords:** pentoxifylline, anti-angiogenesis, tumor, endothelial cells

## Abstract

Angiogenesis is the physiological process of developing new blood vessels to facilitate the delivery of oxygen and nutrients to meet the functional demands of growing tissues. It also plays a vital role in the development of neoplastic disorders. Pentoxifylline (PTX) is a vasoactive synthetic methyl xanthine derivative used for decades to manage chronic occlusive vascular disorders. Recently, it has been proposed that PTX might have an inhibitory effect on the angiogenesis process. Here, we reviewed the modulatory effects of PTX on angiogenesis and its potential benefits in the clinical setting. Twenty-two studies met the inclusion and exclusion criteria. While sixteen studies demonstrated that pentoxifylline had an antiangiogenic effect, four suggested it had a proangiogenic effect, and two other studies showed it did not affect angiogenesis. All studies were either in vivo animal studies or in vitro animal and human cell models. Our findings suggest that pentoxifylline may affect the angiogenic process in experimental models. However, there is insufficient evidence to establish its role as an anti-angiogenesis agent in the clinical setting. These gaps in our knowledge regarding how pentoxifylline is implicated in host-biased metabolically taxing angiogenic switch may be via its adenosine A2BAR G protein-coupled receptor (GPCR) mechanism. GPCR receptors reinforce the importance of research to understand the mechanistic action of these drugs on the body as promising metabolic candidates. The specific mechanisms and details of the effects of pentoxifylline on host metabolism and energy homeostasis remain to be elucidated.

## 1. Introduction

Angiogenesis is the essential physiological process of developing new blood vessels to meet growing tissues’ metabolic and nutritional needs [1]. In oxygen- or nutrient-deprived environments, proangiogenic factors such as VEGFs, FGF-2, PDGF, TNF-α, and IL-6, as well as stem cell factors (SCF) are released, activating the surrounding endothelial cells and triggering different cascading pathways [1,2]. These activated endothelial cells release matrix metalloproteinases resulting in the degradation of the basement membrane. This degradation process allows endothelial tip cells to protrude and migrate toward the source of the angiogenic signal [1].

While angiogenesis is a physiologic process, it also occurs in some pathological conditions, such as neoplastic disorders. In tumor microenvironments, the malignant cells may alter the angiogenesis regulation, resulting in abnormal and rapid vascular growth with subsequent tumor growth, metastasis, and resistance to anticancer therapies [1].

Due to the angiogenesis-altering properties of cancer cells, angiogenesis inhibitors have been investigated as a potential therapeutic option against cancer progression. These angiogenesis inhibitors function mainly by interfering with different signaling pathways in the angiogenesis process. There is debate on the utility of angiogenesis inhibitors as a reliable anticancer therapeutic option. However, due to the dynamic ability of malignant cells to develop drug resistance mechanisms, this approach using angiogenesis inhibitors has shown promising results in certain neoplastic disorders [1]. Therefore, the current research on targeting the tumor microenvironment has focused on anti-angiogenesis therapy, involving drugs that either prevent the formation of new blood vessel supply to the tumor or impair existing blood vessels [3]. The strategic therapeutic approach of antiangiogenic drugs can be either selective targeting of the tumor-associated vasculature, increasing the bioavailability of tumor endothelial cells to systemically administered antiangiogenic drugs, or using metronomic therapy to reduce systemic drug toxicity [4,5]. For example, anti-neoplastic agents are associated with different types of hepatotoxicity. Chemotherapy-induced liver injury can present as hepatitis, steatosis, sinusoidal obstruction syndrome, or chronic parenchymal damage. In wound healing, degranulation is crucial because the growth factors and other mediators that platelets release program damaged tissue for repair. Platelets and white blood cells circulate in inactive forms in a normal healthy state. However, in pathological states, such as an injury involving blood vessels, platelets become activated by contact with components of the extravascular connective tissues exposed at the injury site. Platelets and leukocytes have coordinated and cooperative activities in routine wound healing that limit acute inflammation and trigger tissue repair. Under injury conditions, platelets have two essential functions for survival: (i) to drive clotting to stop the bleeding and (ii) to induce inflammation to initiate healing by releasing growth factors and bioactive molecules that activate acute inflammation and program tissue repair. This process affects the viscoelastic fluid of blood. The viscous component of blood arises primarily through the viscosity of blood plasma, while the elastic component arises from the deformation of the red blood cells. These clinical symptoms are the development of significant risk factors for arterial occlusive disease, such as age, high cholesterol and triglycerides, high blood pressure, diabetes, smoking, and a history of plaque build-up in the arteries. Men are more likely than women to develop arterial occlusive disease.

Pentoxifylline (PTX; 3,7-dimethyl-1-(5-oxohexyl)-3,7-dihydro-1H-purine-2,6-dione) is a synthetic xanthine derivative that acts as a phosphodiesterase inhibitor [6]. It has been commonly used to manage chronic occlusive peripheral vascular disorders in the lower extremities [7]. There are also multiple off-label indications of pentoxifylline in different clinical settings, such as acute coronary syndrome, recurrent cerebral transient ischemic attacks, and congestive heart failure [8,9,10]. A growing body of literature suggests that pentoxifylline can also have a modulatory effect on the angiogenic process. This can be attributed to the overall physiologic effects of pentoxifylline and its metabolites, and their impacts on cellular function and the microenvironment. However, this modulatory effect, whether inhibitory or stimulatory, remains controversial [11]. Nonetheless, several studies have suggested that PTX can modulate the G protein-coupled adenosine receptor function, specifically the Gα-coupled A2A receptor (A2AR). Whether PTX acts directly as an A2AR agonist is unclear, although it can increase the response of A2AR to adenosine [12,13,14,15]. A2AR activation induces adenylyl cyclase activity, increasing intracellular cAMP levels. This exciting observation may explain PTX’s ability to increase intracellular cAMP in a non-specific cyclic-3′,5′-phosphodiesterase (PDE) independent manner. Moreover, validating and characterizing other targets, such as receptor tyrosine receptors (RTK) and pathogen-sensing Toll-like receptors (TLR), may define novel mechanisms for particular pentoxifylline effect that can selectively activate or block such targets with reasonable potency for metabolic health and related diseases. For example, the association of G protein-coupled receptors (GPCR) and RTK signaling––including nerve growth factor receptor Trk and insulin receptors upon ligand binding––is comprehensively reviewed by Pyne and colleagues [16,17,18], Abdulkhalek et al. [19], and Haxho et al. [20]. Onfroy et al. [21] proposed a mechanism dictating biased agonism involving G protein stoichiometry through distinct partitioning of receptor-G protein integration. Here, the expression levels of G proteins influence the biased profiling of agonists and antagonists by affecting different membrane distributions of receptor-G protein populations, in that they determine both their activity and efficacy. It may be that the level of Gα-coupled A2A receptor expression in the naïve state influences the partitioning of not only its G protein, but also the co-expressed receptor in different membrane domains [21]. Here, GPCRs can select more than one active state, called “biased agonism,” “functional selectivity,” or “ligand-directed signaling” [22,23]. Similarly, an array of allosteric ligands can have different degrees of modulation where they facilitate “biased modulation” and can vary dramatically in a probe- and pathway-specific manner [22,24,25]. This biased modulation is not due to differences in orthosteric ligand efficacy or stimulus-response coupling.

In this review, we explore the current evidence regarding the biased modulatory effect of pentoxifylline on the angiogenic process involving RTK and TLR signaling, emphasizing its potential therapeutic utility as an anti-angiogenesis agent in different pathological conditions.

## 2. Study Design and Search Strategy

We conducted an electronic search in the PubMed/MEDLINE and Embase databases from inception to December 2022 for English-language-only literature. In summary, key terms included “Pentoxifylline” and “Angiogenesis or Neogenesis.” A manual search of Google Scholar was also performed to supplement the electronic search. A Queen’s University Medical Science Library librarian was consulted during the literature search. 

All studies that directly assessed the angiogenesis-related properties of pentoxifylline were included. Studies on animal models and human cellular models were also included. The inclusion criteria included primary articles published in English with no date restrictions, while case reports, editorials, and other reviews were excluded. The outcome of interest was the potential interaction between pentoxifylline and the angiogenic process. Two investigators independently reviewed titles, abstracts, and full-text articles against the specified inclusion criteria (W.K. and R.T.) using the Covidence^®^ systematic review screening and data extraction tool (Veritas Health Innovation, Melbourne, Australia). Discrepancies were resolved through consensus. A third reviewer (M.E.-D.) checked the collected data for completeness and accuracy. A study characteristics table was created to summarize the selected articles. Due to significant variability in the format of the reported results, creating a standardized data summary table was impossible. The relevant reported results are discussed in their respective sections.

## 3. Results

The database literature search yielded 155 records, from which five duplicate records were removed. Of the 150 available records, 93 remained eligible after title and abstract screening, thereby proceeding to full-text screening. Full-text screening yielded 22 records that met the eligibility criteria (Figure 1). All of the included studies were laboratory studies that examined the ability of pentoxifylline to generate an effect on angiogenesis in various models. Fifteen studies were in vivo only [26,27,28,29,30,31,32,33,34,35,36,37,38,39,40], three studies were in vitro only [41,42,43], and four studies had both in vivo and in vitro models [44,45,46,47]. The in vitro studies used human cells [42,44,45,46,47] and mouse cell lines [41,43,44,45]. The in vivo studies were mainly conducted using mouse [29,31,34,37,38,45,47] and rat [30,32,33,35,39,40,44] models, but zebrafish embryos [36], rabbit [26,27], and monkey [26,27,28] models were used as well. Study characteristics and their key findings are summarized in Table 1.

In all these studies, pentoxifylline was assessed as influencing angiogenesis in various pathological conditions. Cancer cell models were the most investigated, namely, melanoma [26,27,28,34,45], colon [29], prostate [44], and breast cancer cell models [47]. In addition, one study examined the role of angiogenesis in the context of radiation-induced osteoradionecrosis [40]. Seven studies reported other pathological states, such as hepatopulmonary syndrome [30], peritoneal adhesions [31,33,38], endometriosis [32], bone defects [35], and healing post-skin flap operation [39]. One study examined the embryonic development of zebrafish when exposed to pentoxifylline [36]. Furthermore, two in vitro studies examined macrophage models exposed to pentoxifylline and its effects on angiogenesis factor release [43,44]. Lastly, one in vitro study assessed the effect of pentoxifylline in a novel mouse embryo proepicardium model of angiogenesis and a mouse endothelial cell line [41].

Overall, the included studies did not demonstrate uniformity regarding the effects of pentoxifylline on angiogenesis. Of the 22 included studies, pentoxifylline was found to have antiangiogenic effects in 16 studies [26,27,28,29,30,31,32,34,36,37,38,41,44,45,46,47], proangiogenic effects in 4 studies [35,39,40,43], and no effect at all on angiogenesis in 2 studies [33,42].

## 4. Discussion

Pentoxifylline is a xanthine derivative characterized by the methylation of the first nitrogen of its xanthine ring [6]. Pentoxifylline is administered as an oral drug and is extensively and rapidly absorbed via the gastrointestinal tract, reaching peak plasma concentration in about 15–25 min following oral administration [48]. Pentoxifylline does not significantly bind to plasma proteins and has a relatively uniform body tissue distribution [49]. Erythrocytes and the liver are the main sites of pentoxifylline metabolism [7]. There are seven known metabolites of pentoxifylline, aptly named I-VII, with metabolites I, IV, and V being detectable in the plasma [49]. Pentoxifylline is rapidly eliminated, with most excretion occurring through the urine as metabolite V [49].

Most included studies suggested that pentoxifylline may modulate angiogenesis in animal and cell models representing different neoplastic and pathological conditions. These studies demonstrated the potential effects of pentoxifylline on angiogenesis inhibition, endothelial cell growth inhibition, and decreased tumor volume in different models that included melanoma cells, human tumor implants, rat prostate, and mouse colon cancers [26,27,29,44,45].

To understand the exact anti-angiogenesis mechanism of pentoxifylline, it is essential to highlight that pentoxifylline affects multiple cellular functions resulting in various changes in the cellular microenvironment. More specifically, pentoxifylline can inhibit the production and release of cytokines, including IL-1β, IL-6, IL-8, IL-10, TNF-α, and TNF-β [50]. It can also impair the function of existing TNF-α [51]. In the included studies, pentoxifylline (PTX) was shown to downregulate and, in some cases, inhibit these cytokines, in both in vivo and in vitro models [50,52]. The impact of the PTX is immediate; 6 h after the addition of PTX, a decrease in the production of IL-6 is seen [50]. In addition to the immediate reaction, the response to pentoxifylline by cytokines seems to be dose-dependent. At lower doses (10^−4^ M), the production of only TNFα and IL-8 appears to be inhibited, whereas all cytokine production is inhibited at 10^−3^ M [52]. While the extent to which these cytokines are inhibited differs between cellular/animal models and the dosage of pentoxifylline, there is consistency in the impact of PTX on these cytokines [53]. Moreover, pentoxifylline can modulate immune system responses via its effects on neutrophil degranulation, T and B lymphocyte activation, natural killer cell activity, endothelial leukocyte adhesion, and leukocyte deformability [54].

Studies have shown that PTX can affect the release and function of some predominantly proangiogenic vascular endothelial growth factors [2]. Specifically, pentoxifylline may inhibit the release of the VEGF family of pro-angiogenesis factors (notably VEGF-A and VEGF-C) via their receptors VEGFR1 and VEGFR2 [30,31,32,34,47]. In addition, recent evidence has shown that pentoxifylline inhibits tumor growth and angiogenesis by inhibiting IL-6 secretion and VEGF–VEGFR2 signaling via the STAT3 signaling pathway, which is associated with the non-classical proangiogenic stem cell factor (SCF) [46].

There is also some controversy on whether the antiangiogenic effect of PTX is intrinsic to the drug itself or whether this effect is the product of a more complex interaction between PTX and other elements in the cellular microenvironment. The study by Niderla-Bielińska et al. [41] found that PTX indirectly inhibits angiogenesis in mouse proepicardial explant cultures via decreasing expression of the Notch1 receptor, one of the five mammalian receptors involved in cellular regulation that also plays an integral part in sprouting angiogenesis, and its delta-like 4 (Dll4) ligand. Dll4 is a transmembrane ligand for Notch receptors expressed in arterial blood vessels and sprouting endothelial cells. However, when the same protocol was conducted on a mouse endothelial cell line, it was found that PTX exhibited no effect on angiogenesis. The authors theorized that the presence of mesenchymal cells is essential to support the antiangiogenic effect of PTX [41]. Similarly, in the study by Ching et al. [29], it was found that PTX could only exert a meaningful antiangiogenic effect in synergy with other drugs, namely, dimethylxanthenone-4-acetic acid (DMXAA). This suggests that other extrinsic factors may be essential to express the inhibitory effect of PTX on angiogenesis. As for the mechanism of action of the synergy between PTX and other anti-tumor agents in inhibiting angiogenesis, this was likely attributed to the similarities in their chemical properties [29].

There is also mixed evidence on the effect of PTX on suppressing the metalloproteinase (MMP) pathways, which are known to be potent triggers of the neo-angiogenic process [55]. Arsenyan et al. [28] showed that PTX did not significantly suppress the MMP-mediated endothelial invasion in their study on the human umbilical vein endothelial cell tube formation model. This contrasts with the findings of Seo et al. [43], which show that MMP-2 was upregulated in the 24 h post administration of pentoxifylline yet down-regulated after 48 h in a mouse-derived macrophage cell line. These findings suggest that metalloproteinase levels may be transiently affected by pentoxifylline.

PTX is a methylxanthine derivative. Methylxanthines are known to be inhibitors of adenosine receptors through the disruption of adenosine signaling and the downregulation of VEGF, thereby inhibiting angiogenesis [56]. However, the exact process by which PTX affects angiogenesis remains poorly understood.

One suggested mechanism is the canonical GPCR “A2BAR”, a biased adenosine receptor that senses adenosine and couples to G proteins (G_s_, G_q_), leading to increases in intracellular cAMP, protein kinase A (PKA), mitogen-activated protein kinase (MAPK), and the protein kinase B (PKB/Akt) signaling axes [57,58]. Adenosine is recognized by four GPCRs: A1AR, A2AAR, A2BAR, and A3AR, and they have been previously characterized by Fredholm et al. [59]. The human A2AAR receptor is a 412-amino-acid-long protein encoded by the gene at chromosome position 22q11.23. The A2B adenosine receptor (A2BAR) is found in many different cell types and requires higher adenosine concentrations for activation than the A1, A2A, and A3 AR subtypes [59]. Thus, unlike the other AR subtypes, the A2BAR is not stimulated by physiological levels of adenosine but may play an essential role in pathophysiological conditions associated with massive adenosine release. Such conditions occur in ischemia or in tumors where hypoxia is commonly observed. Fishman et al. [60] provided an eloquent review of the most exciting effects for a potential anticancer treatment based on A2BARs as a target involving inhibition of angiogenesis and ERK phosphorylation. However, the dilemma is that inhibition of angiogenesis requires A2BAR antagonists, whereas inhibition of growth signaling via the ERK/MAP kinase pathway might be achieved through treatment with A2BAR agonists. Moreover, adenosine signaling via A1AR, A2AAR, A2BAR, and A3AR has been acknowledged as a crucial regulator of the interactions between tumor cells and the tumor microenvironment (TME). High TME concentrations have led to adenosine-initiated signaling pathways contributing to tumor growth. For example, AR can initiate sustained neoangiogenesis, eliciting at the same time, an immunosuppressive environment that hinders immune-based tumor control approaches [61]. For GPCR targeting in cancer, only a few examples have been successfully used as targets to develop drugs that can block cancer-associated pathways.

However, when growth factors bind to their receptor tyrosine kinases (RTK), the GPCR signaling can be potentiated through crosstalk between neuraminidase-1 (Neu-1) and metalloproteinase-9 (MMP-9) as previously described by Gilmour et al. [62]. This crosstalk leads to a conformational change of the receptor, mediating the signaling of ERK/MAPK through zinc-finger factor Snail. Haxho et al. [63] proposed a graphical illustration (Figure 2) depicting a Snail-MMP9 signaling axis [64], maintaining several important cancer growth factor receptor signaling platforms in promoting Neu1-MMP9 crosstalk in complex with glycosylated receptors. Snail has previously been shown to induce the transcription and expression of matrix metalloproteinase-9 (MMP-9) and is linked to epithelial–mesenchymal transition (EMT) and neovascularization in the tumor microenvironment (TME) [64]. Snail, a transcriptional factor and repressor of E-cadherin, is also well known for its role in cellular invasion. It can regulate epithelial to mesenchymal transition (EMT) during embryonic development and in epithelial cells as well as mediate tumor progression and metastases.

Abdulkhalek et al. [64] provided a proof of concept for the role of Snail in tumor progression and metastases in Figure 2. The silencing of Snail and its associate member Slug in human A2780 ovarian epithelial carcinoma cell line was investigated to identify its role in tumor neovascularization. Silencing Snail in A2780 cells abrogated the Neu1 activity following EGF stimulation of the cells compared with A2780 and A2780 Slug knock-down (KD) cells, as depicted in Figure 2. Oseltamivir phosphate (OP) treatment of A2780 and cisplatin-resistant A2780cis cells reproducibly and dose-dependently abated the cell viability with an LD50 of 7 and 4 μM, respectively, after 48 h of incubation. OP is an antiviral medication that blocks the actions of influenza virus types A and B in the body. It has been shown to specifically target neuraminidase-1 (Neu1), depicted in Figure 2 [62]. Heterotopic xenografts of A2780 and A2780 Slug KD tumors developed robust and bloody tumor vascularization in RAG2×Cγ double mutant mice. OP treatment at 50 mg/kg daily intraperitoneally did not significantly impede the A2780 tumor growth rate but did cause a significant reduction of lung metastases compared with the untreated and OP 30 mg/kg cohorts. Silencing Snail in A2780 tumor cells completely abrogated tumor vascularization, tumor growth, and spread to the lungs in RAGxCγ double mutant mice. A2780 and A2780 Slug KD tumors expressed high levels of human EMT markers, N- and VE-cadherins, and host CD31^+^ endothelial cells, while A2780 Snail KD tumors expressed E-cadherin and reduced host CD31^+^ cells. OP 50 mg/kg cohort tumors had reduced numbers of host CD31^+^ cells compared with a higher expression level of CD31^+^ cells in tumors from the untreated control and OP 30 mg/kg cohorts.

Abdulkhalek et al. [19,65] uncovered a molecular organizational GPCR signaling platform to potentiate Neu1 and MMP-9 crosstalk on the cell surface, which is essential for transactivating TLR receptors and subsequent cellular signaling. The GPCR agonists, bombesin, bradykinin, lysophosphatidic acid (LPA), cholesterol, angiotensin-1 and -2, but not thrombin, induce Neu1 activity in live macrophage cell lines and primary bone marrow macrophage cells from wild-type (WT) mice but not from Neu1-deficient mice. Using immunocytochemistry and NFκB-dependent secretory alkaline phosphatase (SEAP) analyses, bombesin induced NFκB activation in BMC-2 and RAW-blue macrophage cells, inhibited by MyD88 homodimerization inhibitor, OP, galardin, piperazine, and anti-MMP-9 antibodies [65]. The bombesin receptor, neuromedin B (NMBR), forms a complex with TLR4 and MMP9. Silencing MMP9 mRNA using siRNA transfection of RAW-blue macrophage cells markedly reduced Neu1 activity associated with bombesin-, bradykinin-, and LPA-treated cells to the untreated controls.

Therefore, the mechanism by which pentoxifylline inhibits angiogenesis may be through the binding to the adenosine receptor A2BAR. This dictates biased agonism involving G protein stoichiometry through the different partitioning of TLR receptor-G protein integration (Figure 3). Here, the expression levels of G proteins would influence the biased profiling of agonists and antagonists by affecting different membrane distributions of TLR-G protein populations in that they determine both their activity and efficacy. It may be that the level of Gα-coupled A2A receptor expression in the naïve state influences the partitioning of not only its G protein, but also the co-expressed receptor in different membrane domains [21]. Here, GPCRs can select more than one active state called “biased agonism,” “functional selectivity,” or “ligand-directed signaling” [22,23], leading to Neu1-MMP9 crosstalk and Snail signaling (Figure 3). For instance, p38 and related kinases can act as cofactors in NF-κB activation, and significant overlap exists between the stimuli that activate NF-κB and the stimuli that activate MAPKs [66].

Interestingly, NF-κB signaling has yet to be studied in pentoxifylline-mediated epigenetic effects, even though its association with MAPKs and its ability to induce epigenetic modifications support its involvement [66]. The NF-κB family consists of five structurally related transcription factors that form homodimersand heterodimers with one another [67]. Inactivated NF-κB dimers are located in the cytoplasm in association with inhibitory proteins called inhibitors of kB (IkB) [68]. A plethora of factors can induce the phosphorylation and subsequent ubiquitin-dependent degradation of IkB, enabling the translocation of its associated NF-κB subunits to the nucleus, where they can regulate the transcription of many genes [68]. The dilemma is that inhibition of angiogenesis requires A2BAR antagonists, whereas inhibition of growth signaling via the ERK/MAP kinase pathway might be achieved through treatment with A2BAR agonists. It is hypothesized that the level of Gα-coupled A2A receptor expression in the naïve state influences the partitioning of its G protein and the co-expressed receptor in different membrane domains, staging the NF-κB signaling in pentoxifylline-mediated epigenetic effects (Figure 3).

These pathways can also play an essential role downstream of the receptor in multiple mechanisms, one of which is supporting the synthesis of the inflammatory and profibrotic cytokines, TGF-β and IL-6 [57]. Adenosine GPCR A2BAR is also highly expressed in cardiac fibroblasts and endothelial cells, where it is hypothesized to have a cardioprotective role through the regulation of myocardial remodeling [57]. Moreover, it is found in most inflammatory cells where it induces the secretion of IL-6 from macrophages, IL-1β, IL-8, IL-4, and VEGF from mast cells, and TNF-α from bronchial epithelial cells [58]. In contrast, the adenosine receptor also meditates anti-inflammatory effects by inhibiting the production of TNF-α and IL-1β in neutrophils [58].

Of note, it is suggested that the anti-inflammatory characteristics of PTX were mainly attributed to its inhibitory effect on TNF and other inflammatory cytokines. However, none of these studies specifically examined the direct effect of PTX on the angiogenesis process. For example, PTX has shown promising results in treating a broad spectrum of inflammatory and neoplastic conditions in the clinical setting. The efficacy of PTX was studied in the treatment of several inflammatory conditions, such as Crohn’s disease [69], acute pancreatitis [70], and vitiligo [71]. Similarly, other studies have investigated pentoxifylline’s role in preventing irradiation-induced gastrointestinal inflammatory side effects [72] and pulmonary toxicity [73]. In addition, multiple clinical trials have explored the role of pentoxifylline as an adjunct treatment for some neoplastic conditions such as colon cancer [74], lung cancer [75], and squamous cell carcinoma of the head and neck [76].

However, multiple other studies suggest that PTX may exhibit a proangiogenic effect. In the study of Çakmak and colleagues, PTX resulted in enhanced angiogenesis during the healing process of segmental cortical bone defects of the radius bone in a rat model [35]. This can be potentially explained by the fact that PTX can improve microcirculatory flow and tissue perfusion by decreasing blood viscosity and increasing red blood cell deformability [77], decreasing fibrinogen, *α*_2_-antiplasmin, *α*_1_-antitrypsin, *α*_2_-macroglobulin, and platelet aggregation and adhesion [54]. It also increases plasminogen activator, plasmin, and antithrombin III, thereby resulting in an overall enhancement of tissue perfusion and healing [54]. Pentoxifylline may increase the expression of the proangiogenic factors, PECAM and VEGF-A, only in combination with tocopherol in a rat model with radiation-induced osteoradionecrosis [40]. However, TNF-α was decreased with PTX combined with tocopherol but increased when only PTX was used in this rat model [40]. Another study reported that PTX might increase the proangiogenic factors, HIF, angiogenin, and VEGF-C expression in a mouse cell line derived from macrophages [43]. However, neither of these studies identified the underlying mechanism that may explain these findings. In their study of rats that underwent a surgical skin flap procedure, Pedretti and colleagues showed that pentoxifylline was an effective angiogenic factor when delivered subcutaneously [39]. The authors reported that the levels of VEGF were elevated while the levels of TGF-β1, a known angiogenic inhibitory factor [78], were reduced locally in the endothelial cells. This suggests that PTX might be activated differently depending on the route of administration, as this study was the only one of the included studies to have administered pentoxifylline subcutaneously rather than orally or by direct exposure of cells to pentoxifylline solution [39].

Boztosun and colleagues attributed the lack of effect of PTX on angiogenesis to the fact that their model of surgically induced adhesions resulted in ischemia in tissues rather than the bleeding that other adhesion protocols caused, which may have interfered with the effects of pentoxifylline [33]. These findings align with the previously mentioned concept of the interaction between PTX and other elements in the surrounding cellular microenvironment.

Finally, it is essential to highlight that PTX may be associated with multiple untoward side effects, such as mild headaches, pain in the extremities, and gastrointestinal symptoms [7]. These can be explained by its vasoactive chemical properties and complex interaction with various immunological cells and pathways [7]. Therefore, monitoring for any unexpected physiological or pathological consequences associated with its use in the clinical setting will be necessary.

**Figure 3 cells-12-01199-f003:**
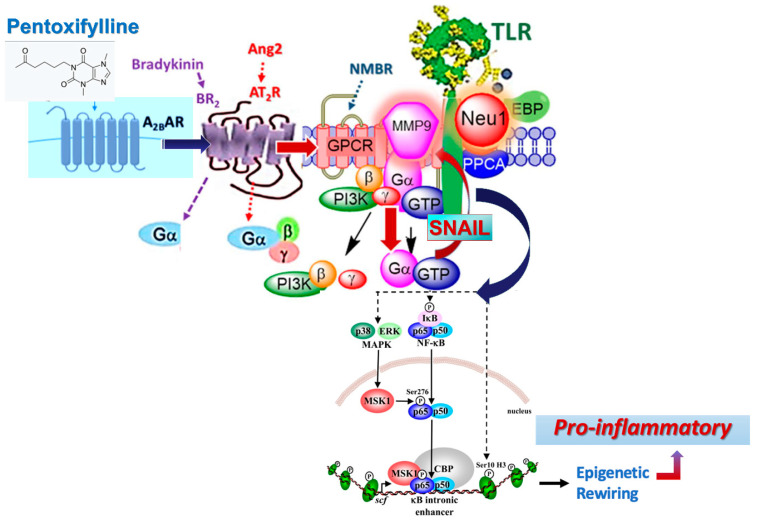
Pentoxifylline binds to A2BAR to induce downstream Snail signaling. Bradykinin (BR2) and angiotensin II receptor type I (AT2R) are tethered within a multimeric receptor Toll-like receptor (TLR) transactivation signaling axis, mediated by Neu1 sialidase and the glycosylation modification of TLRs (adapted from Abdulkhalek et al., 2012 [62]). Here, the link regulating the interaction of these molecules and their signaling mechanism(s) on the cell surface reveals a novel biased GPCR signaling process in inducing the TLR transactivation signaling, with subsequent activation of Neu1 sialidase and the modification of the receptor glycosylation. The biased GPCR signaling platform potentiates Neu1 and MMP-9 crosstalk on the cell surface, essential for transactivating TLRs and subsequent NFκB cellular signaling and inducing epigenetic rewiring. Notes: TLR ligand and GPCR agonists can potentiate biased NMBR-TLR signaling and induce MMP-9 activation and Neu1 sialidase activity. Activated MMP-9 is proposed here to remove the EBP as part of the molecular multi-enzymatic complex that contains β-galactosidase/Neu1 and PPCA. Activated Neu1 then hydrolyzes α-2,3 sialyl residues of TLR at the ectodomain to remove steric hindrance to facilitate TLR association and subsequent recruitment of MyD88 and downstream signaling. Citation: Taken in part from Qorri et al. [79] Cells 2018, 7(9), 117; https://doi.org/10.3390/cells7090117 © 2023 Qorri et al. and Jakowiecki et al. [80] Molecules 2021, 26, 2456. https://doi.org/10.3390/molecules26092456. © 2023 by the authors. Licensee MDPI, Basel, Switzerland. This article is an open access article distributed under the terms and conditions of the Creative Commons Attribution (CC BY) license (http://creativecommons.org/licenses/by/4.0/ (accessed on 23 April 2021), and Reber et al. [81] PLoS ONE 2009; 4(2): e4393. Published online 6 February 2009. https://doi.org/10.1371/journal.pone.0004393. © 2023 Reber et al. This open access article is distributed under the terms of the Creative Commons Attribution License, which permits unrestricted use, distribution, and reproduction in any medium, provided the original author and source are properly credited.

## 5. Limitations of the Study

First, none of the included studies were conducted on human subjects, making it challenging to extrapolate their findings into the clinical setting. Second, there was no consistency between the included studies regarding their design and methodology. Third, the process of angiogenesis is very complex and entails multiple interacting cellular and humoral pathways; therefore, it may be challenging to identify the isolated effects of pentoxifylline on this process. Finally, the contradictory findings of these studies may be attributable to the lack of identification of possible interactions between pentoxifylline and other drugs or elements in the surrounding cellular microenvironment.

## 6. Conclusions

The current evidence suggests that pentoxifylline may play a modulatory role in angiogenesis. However, these findings are derived from studies conducted on animal and cell models. Further research is required to establish the role of PTX in angiogenesis in the clinical setting in the context of different inflammatory and neoplastic disorders.

## 7. Future Research Directions and Limitations

The gaps in knowledge regarding how pentoxifylline is implicated in host angiogenic switch metabolically taxing via adenosine A2BAR, GPCR, and GPCR receptors highlight the importance of further research to understand the mechanistic action of these drugs on the body as promising metabolic candidates. The specific mechanisms and details of the effects of PTX on the host metabolism and energy homeostasis remain to be elucidated. Future research in PTX pharmacology will likely focus on the following areas.

Exploring the structure–activity relationships of PTX that target the adenosine GPCR site or behave as neutral A2B adenosine receptor (A2BAR) antagonists.Assessing the therapeutic potential of pentoxifylline-biased A2BAR receptor allosteric modulators and neutral antagonists.Validating and characterizing other targets, such as RTK and TLR receptors, for particular PTX effects and developing compounds in selectively activating or blocking targets with potency for metabolic health and related diseases.Validating and characterizing A2B adenosine receptors as homodimers, heterodimers, or oligomers with one or more classes of other receptors.Validating and characterizing the angiogenesis system in ameliorating the symptoms or the underlying pathology of certain disorders.

## Figures and Tables

**Figure 1 cells-12-01199-f001:**
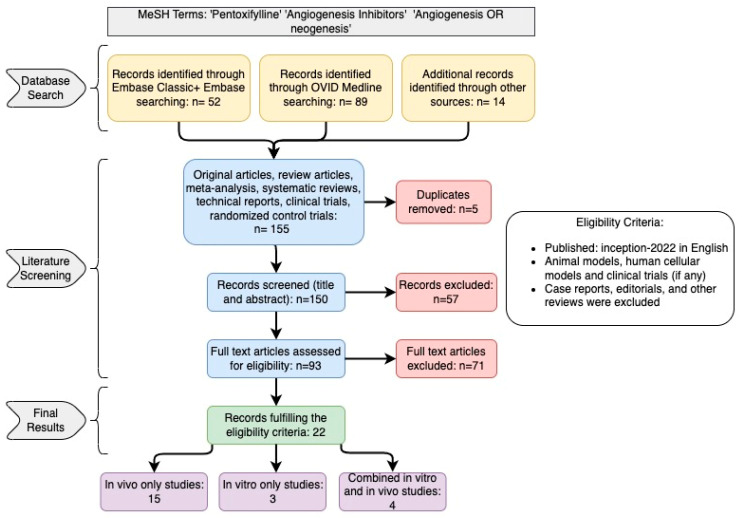
Literature Search and Screening Flowchart. The database search results (yellow) were compiled into a pool of eligible records that would proceed through the literature screen (blue). Duplicates and ineligible records were excluded (red). The remaining records that met the eligibility criteria were included in the review and broken down by study design (purple).

**Figure 2 cells-12-01199-f002:**
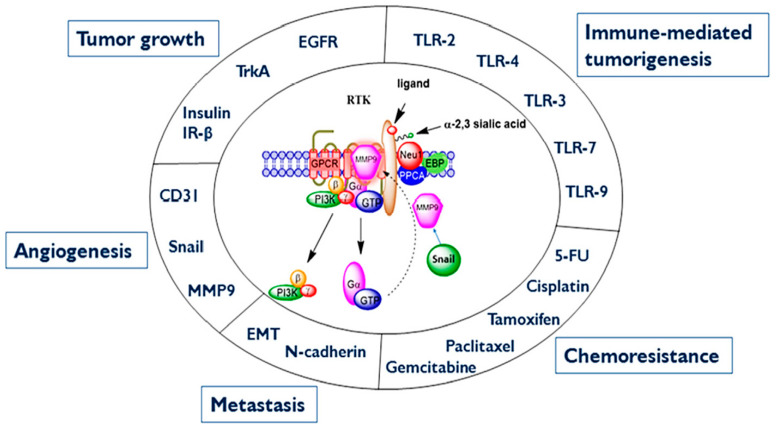
SNAIL and MMP-9 signaling axis in facilitating a neuraminidase-1 (Neu1) and matrix metalloproteinase-9 (MMP-9) crosstalk in regulating receptor tyrosine kinases (RTKs) in cancer cells to promote tumor neovascularization. Notes: Snail and MMP-9 expressions are closely connected in similar invasive tumor processes for cancer. Snail induces MMP-9 secretion via multiple signaling pathways, but particularly in cooperation with oncogenic H-Ras (RasV12), Snail leads to the transcriptional up-regulation of MMP-9. This Snail–MMP-9 signaling axis is the connecting link in promoting growth factor receptor glycosylation modification involving the subsequent receptor signaling platform of a Neu1-MMP-9 crosstalk tethered at the ectodomain of RTKs. Activated MMP-9 removes the elastin-binding protein (EBP) as part of the molecular multi-enzymatic complex that contains β-galactosidase/Neu1 and protective protein cathepsin A (PPCA). Activated Neu1 hydrolyzes α-2,3-sialic acid residues of RTKs at the ectodomain to remove steric hindrance to receptor association and activation. This process sets the stage for SNAIL’s role in tumor neovascularization. Abbreviations: GPCR, G protein-coupled receptor; Pi3K, phosphatidylinositol 3-kinase; GTP, guanine triphosphate; EBP, elastin-binding protein; PPCA, protective protein cathepsin A. Citation: [19] Taken in part from Research and Reports in Biochemistry 2013:3 17–30. © 2013 Abdulkhalek et al., publisher and licensee Dove Medical Press Ltd. This is an open access article that permits unrestricted non-commercial use, provided the original work is properly cited.

**Table 1 cells-12-01199-t001:** Summary of the included studies.

Study [Reference]	Study Design (In Vivo)	Study Design (In Vitro)	Findings
Ambrus et al., 1991 [26]	Human malignant melanoma implants in rabbit cornea and non-human primates.		Pentoxifylline inhibited human tumor implant-induced angiogenesis.
Ambrus et al., 1992 [27]	Human malignant melanoma implants in rabbit cornea and non-human primates.		Pentoxifylline inhibited human tumor implant-induced angiogenesis.
Ching et al., 1998 [29]	Interaction of thalidomide, phthalimide analogs of thalidomide and pentoxifylline with anti-tumor agent DMXAA in mice models given Colon 38 tumors.		Pentoxifylline potentiates DMXAA inhibition of serum TNF production.
Joseph & Isaacs, 1998 [44]	Transplanted Dunning R-3327 MAT-Lu rat prostate cancers. Pentoxifylline effects were assessed on tumor-associated macrophages and blood vessel densities.	On human and mouse macrophages.	Pentoxifylline inhibition of macrophage secretion tumor necrosis factor-α and granulocyte–macrophage colony-stimulating factor, and reduction of tumor blood vessel density and tumor growth.
Ambrus et al., 2000 [28]	Human malignant melanoma implants in *Macaca arctoides* monkey cornea.		Pentoxifylline inhibited human tumor implant-induced angiogenesis.
Gude et al., 2001 [45]	C57BL/6J mice model injected with B16-F10 melanoma cells. IC50 7 mM after 24 h exposure.	Growth inhibition of 2 endothelial cell lines.	Significant inhibition of tumor-induced angiogenesis in C57B1/6 mice inoculated with pentoxifylline that paralleled decreased tumor volumes. In vitro pentoxifylline exhibited a dose-response inhibition of endothelial cell growth and downregulation of urokinase-type plasminogen activator expression.
Zhang et al., 2009 [30]	Hepatopulmonary syndrome induced in a rat model; pulmonary angiogenesis was assessed by quantifying factor VIII-positive micro vessels and levels of von Willebrand factor, vascular endothelial cadherin, angiogenic factors, and proliferating cell nuclear antigen.		Pentoxifylline-treated rats had a reduction in micro vessels and lung monocyte accumulation, downregulation of pulmonary angiogenic factors, and decreased symptoms of hepatopulmonary syndrome. The authors conclude that pentoxifylline decreases hepatopulmonary syndrome-associated angiogenesis, decreases the associated symptoms, and downregulates VEGF-A mediated pathways.
Mendes et al., 2009 [31]	Murine model of sponge-induced peritoneal adhesion, treated with pentoxifylline and assessed by measuring hemoglobin content, VEGF, and morphometric analysis.		Following treatments of pentoxifylline, hemoglobin content, morphometric, morphometric analysis of vessel number, and levels of VEGF decreased. The results align with previous evidence that anti-VEGF activity is associated with angiogenesis inhibition.
Vlahos et al., 2010 [32]	Surgical induction of endometriosis in rats. Morphological changes and VEGF-C and FLK-1 expression were assessed.		There was a significant reduction in the mean volume of endometriotic implants in the pentoxifylline treatment groups. There was a significant reduction in VEGF-C and FLK-1 expressions. The authors conclude that pentoxifylline may suppress angiogenesis by downregulating VEGF-C and FLK-1 expression.
Boztosun et al., 2012 [33]	Surgical induction of adhesions in rats. Morphological changes and VEGF, bFGF, TGF-β, and PDGF expression were assessed.		Pentoxifylline did not show any effect on the expression of angiogenic factors.
Pratibha et al., 2013 [34]	This study investigated the mechanisms for the antiangiogenic activity of pentoxifylline by injecting B16-F10 melanoma cells into C57BL/6 mice and assessing blood vessel density and molecular markers. IC50 39.2 +/− 1.3 mM after 2 h exposure.		The results of this study demonstrated that pentoxifylline: suppressed STAT3 phosphorylation and its upstage kinases, reduced expression of HIF1α, VEGF, VEGFR1, VEGFR2, and pro-inflammatory cytokines, and suppressed tumor volume and micro vessel density. The authors conclude that pentoxifylline may exert anti-tumor activity by inhibiting angiogenesis through the STAT3 pathway in B16F10 melanoma.
Kamran & Gude, 2013 [46]	Intra-dermal mouse xenograft model was used to assess tumor volume and angiogenesis. IC50 7 mM after 2 h exposure with pentoxifylline.	A375 human melanoma cell line was treated with pentoxifylline and assessed for STAT3 signaling.	Following treatment of the mice with pentoxifylline, there was a significant decrease in the mean volume of the tumors and a reduction in tumor-induced angiogenesis. Pentoxifylline’s tumor growth and angiogenesis inhibition may involve the STAT3 signaling pathways.
Nidhyanandan et al., 2015 [47]	MS-275 and pentoxifylline were assessed in a murine Matrigel plug angiogenesis model and human breast cancer (MDA-MB-231) xenograft model.	A panel of cancer cell lines was treated with pentoxifylline and MS-275 and evaluated for cellular proliferation, cell cycle regulation, apoptosis, and anti-angiogenesis.	A combination of MS-275 and pentoxifylline significantly inhibited angiogenesis in the Matrigel plug angiogenesis assay. The combination therapy inhibited the expression of VEGF in a dose-dependent manner.
Çakmak et al., 2015 [35]	Sprague–Dawley rats were utilized to determine the effect of pentoxifylline on angiogenesis and bone healing. Radiographic, immunohistochemical methods and histological methods were utilized to evaluate the effect.		Pentoxifylline may improve angiogenesis and healing of segmental cortical bone defects of the radius in a rat model.
Nathan et al., 2016 [36]	Various concentrations of pentoxifylline were tested at 50% epiboly stage (5.2 HPF) of zebrafish embryos and evaluated phenotypic changes and expression of adenosine receptors, HIF-1α, VEGFaa, VEGFr2, and RP-1a.		RBC staining demonstrated an absence of intersegmental vessels in embryos treated with pentoxifylline. Pentoxifylline-treated embryos developed abnormal vasculature. Additional results show inhibition of VEGFAA and adenosine receptions and new blood vessel formation following treatment with pentoxifylline.
Bałan et al., 2017 [37]	Tumor cells were incubated with various concentrations of pentoxifylline before transplantation into mice.		The results of this study demonstrate that pentoxifylline had an inhibitory effect on tumor growth and volume and had a dose-dependent decrease in angiogenesis following transplantation.
Niderla-Bielińska et al., 2018 [41]		Mouse embryo proepicardium was harvested and treated with pentoxifylline to assess the expression of angiogenic factors. Endothelial cell line C166 was derived from embryonic yolk sac treated with pentoxifylline to assess the direct effect on angiogenesis.	Pentoxifylline indirectly inhibits angiogenesis in mouse proepicardial explant cultures by decreasing Dll4 and Notch1 expression but has no significant effect on the C166 endothelial cell line.
Yang et al., 2018 [38]	A mouse model was used to investigate pentoxifylline’s effect on postoperative intra-abdominal adhesion formation through angiogenesis and other physiological processes. Angiogenesis was assessed via immunohistology analysis of angiogenesis markers Ki67^+^/CD31^+^.		Pentoxifylline significantly suppressed angiogenesis during the peritoneal repair of the mice. The authors state that these findings are in line with additional studies on the inhibition of angiogenesis.
Arsenyan et al., 2020 [42]		Matrigel (BD Biosciences) human umbilical vein endothelial cell tube formation model was used to investigate various compounds’ angiogenesis and MMP inhibition activity.	The study results show that pentoxifylline did not inhibit any of the studied matrix metalloproteinases. The authors presume that the lack of correlations between MMP and angiogenesis inhibition indicates that the compounds modulate angiogenesis via different mechanisms.
Pedretti et al., 2020 [39]	Rat model of skin flap surgical procedure, then treated with subcutaneous pentoxifylline. VEGF and TGF-β1 levels were measured.		Pentoxifylline stimulated angiogenesis and reepithelization while reducing fibrogenesis.
Seo et al., 2020 [40]	Rat model of radiation-induced osteoradionecrosis. Treated with pentoxifylline alone and with pentoxifylline and tocopherol. Angiogenesis effects were assessed.		Pentoxifylline and tocopherol work synergistically to promote angiogenesis, while pentoxifylline alone had a slight increase in proangiogenic factors PECAM, VEGF-A, and TNF-α.
Seo et al., 2021 [43]		The effects of pentoxifylline on RAW 264.7 cells were analyzed with immunoprecipitation high-performance liquid chromatography to assess angiogenesis inhibition via the expression levels of VEGF-A, vWF, ET-1, CD31, MMP-10, and VCAM.	The expression levels of VEGF-A, vWF, ET-1, CD31, MMP-10, and VCAM showed a minimal change within 5%. The results of this study show that pentoxifylline has a weak angiogenic effect over 48 h.

Abbreviations: C57BL/6J: C57 black 6J; CD31: Cluster of differentiation 31; C166: Cellosaurus 166; Dll4: Delta-like 4; DMXAA: 5,6-dimethylxanthenone-4-acetic acid; ET-1: Endothelin 1; FGF: Fibroblast growth factor; Flk-1: Fetal Liver Kinase-1; MDA-MB-231: M.D. Anderson—Metastatic Breast 231; MMP: matrix metalloproteinase; MS-275: Entinostat; Notch1: Neurogenic locus notch homolog protein 1; NRP-1a: neuropilin 1a; PECAM: Platelet endothelial cell adhesion molecule, PDGF: Platelet-derived growth factor; RAW 264.7: Ralph Raschke Watson cell line 264.7; R-3327 MAT-Lu: R-3327 metastatic, anaplastic tumor to the lung; STAT3: Signal transducer and activator of transcription 3; TGF: tumor growth factor; TNF: tumor necrosis factor; VCAM: vascular cell adhesion molecule; VEGF: Vascular endothelial growth factor; VEGFR: Vascular endothelial growth factor receptor; vWF: von Willibrand factor.

## Data Availability

Data are contained within the article.

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
