# Peer review of "Parsimonious Effect of Pentoxifylline on Angiogenesis: A Novel Pentoxifylline-Biased Adenosine G Protein-Coupled Receptor Signaling Platform"

_cells, 2023, doi:10.3390/cells12081199_

Round 1

Reviewer 1 Report

Overall, the paper seemed to be grounded in good science. There does not seem to be any bias or self-promotion, and the authors did a good job of not being too hyperbolic in the speculation of the effect of Pentoxifylline on angiogenesis in therapeutics.  I have some comment.

1.      Does Pentoxifylline have different effect on angiogenesis in different type of tumors?

2.      Are there any related clinical trials of Pentoxifylline? The clinical trials may provide the direct evidence of clinical settings of the therapeutic potential of Pentoxifylline.

3.      The information of Adenosine G Protein-Coupled Re- 3 ceptor Signaling in cancer should be given in Introduction.

4.      The function of Pentoxifylline binds to A2BAR in cancer seems to be of extra significance. Perhaps this paragraph can be expanded to include more information at hand.

5.      (Dll4) expand on first use.

Author Response

Reviewer #1:

Comments and Suggestions for Authors

Overall, the paper seemed to be grounded in good science. There does not seem to be any bias or self-promotion, and the authors did a good job of not being too hyperbolic in the speculation of the effect of Pentoxifylline on angiogenesis in therapeutics.  I have some comment.

  1. Does Pentoxifylline have different effect on angiogenesis in different type of tumors?

 Author responseThank you for this comment. Regarding the tumour type, the findings in the literature are inconsistent, and there is no clear pattern to suggest that pentoxifylline has different effects on different tumour cells.  However, the studies that showed pro-angiogenesis effects of pentoxifylline were mainly performed on non-tumour cells (21,25,26,29):

  1. Çakmak, G.; Şahin, M.; Özdemİr, B.H.; Karadenİz, E. Effect of pentoxifylline on healing of segmental bone                    defects and angiogenesis. Acta Orthop Traumatol Turc 2015, 49, 676-682, doi:10.3944/aott.2015.15.0158.
  2. Pedretti, S.; Rena, C.L.; Orellano, L.A.A.; Lazari, M.G.; Campos, P.P.; Nunes, T.A. Benefits of pentoxifylline for skin flap tissue repair in rats. Acta Cir Bras 2020, 35, e301105, doi:10.1590/acb351105.
  3. Seo, M.H.; Myoung, H.; Lee, J.H.; Yang, H.C.; Woo, K.M.; Lee, S.K.; Kim, S.M. Effects of pentoxifylline and tocopherol on an osteoradionecrosis animal model. J Craniomaxillofac Surg 2020, 48, 621-631, doi:10.1016/j.jcms.2020.02.008.
  4. Seo, M.H.E., M.Y.; Nguyen, T.T.H.; Yang, H.J.; Kim, S.M. . Immunomodulatory Effects of Pentoxifylline: Profiling Data Based on RAW 264.7 Cellular Signaling. Applied Sciences 2021, 11, doi:covidwho-1390523.
  5. Are there any related clinical trials of Pentoxifylline? The clinical trials may provide the direct evidence of clinical settings of the therapeutic potential of Pentoxifylline.

 Author response:  Thank you for this comment. There are several clinical trials that studied the anti-inflammatory characteristics of pentoxifylline (mainly via its inhibitory effect on TNF and other cytokines) but did not look specifically at its anti-angiogenesis effects.  We have added a section that discusses some of the most recent clinical trials that investigated the role of pentoxifylline in the context of several inflammatory and neoplastic conditions. “Of note, it is suggested that the anti-inflammatory characteristics of pentoxifylline were mainly attributed to its inhibitory effect on TNF and other inflammatory cytokines. However, none of these studies specifically examined the direct effect of pentoxifylline on the angiogenesis process. For example, pentoxifylline has shown promising results in treating a broad spectrum of inflammatory and neoplastic conditions in the clinical setting. The efficacy of pentoxifylline was studied in the treatment of several inflammatory conditions, such as Crohn’s disease [69], acute pancreatitis [70], and vitiligo [71]. Similarly, other studies investigated pentoxifylline’s role in preventing irradiation-induced gastrointestinal inflammatory side effects [72] and pulmonary toxicity [73]. In addition, multiple clinical trials have explored the role of pentoxifylline as an adjunct treatment for some neoplastic conditions such as colon cancer [74], lung cancer [75] and squamous cell carcinoma of the head and neck [76].”

  1. The information of Adenosine G Protein-Coupled Receptor Signaling in cancer should be given in Introduction.

Author responseThank you for this comment. We have included information on how adenosine G Protein-coupled receptor signaling is regulated via partitioning. “Several studies have suggested that PTX can modulate G protein-coupled adenosine receptor function, specifically the Gα-coupled A2A receptor (A2AR). Whether PTX acts directly as an A2AR agonist is unclear, although it can clearly increase the response of A2AR to adenosine [12-15]. A2AR activation induces adenylyl cyclase activity, which in turn increases intracellular cAMP levels. This interesting observation may explain PTX's ability to increase intracellular cAMP in a non-specific cyclic-3',5'-phosphodiesterase (PDE) independent manner. Also, validating and characterizing other targets, such as receptor tyrosine receptors (RTK) and pathogen-sensing TOLL-like receptors (TLR), may de-fine novel mechanisms for pentoxifylline effect that can selectively activate or block such targets with reasonable potency for metabolic health and related diseases. For example, it is well known that the association of G protein-coupled receptors (GPCR) and RTK signaling including nerve growth factor receptor Trk and insulin receptors upon ligand binding is eloquently reviewed by Pyne and colleagues [16-18], Abdulkhalek et al. [19], and Haxho et al. [20]. Onfroy et al. [21] proposed a mechanism dictating biased agonism involving G protein stoichiometry through distinct partitioning of receptor-G protein integration. Here, the expression levels of G proteins influence the biased profiling of agonists and antagonists by affecting different membrane distribution of receptor-G protein populations, in that they determine both their activity and efficacy. It may be that the level of Gα-coupled A2A receptor expression in the naïve state influences the partitioning of not only its G protein but also the coexpressed receptor in different membrane domains [21]. Here, GPCRs can select more than one active state that is called ‘biased agonism’, ‘functional selectivity’, or ‘ligand-directed signaling’ [22,23]. Similarly, an array of allosteric ligands can have different degrees of modulation where they facilitate ‘biased modulation’ and can vary dramatically in a probe- and pathway-specific manner [22,24,25]. This biased modulation is not due to differences in orthosteric ligand efficacy or stimulus-response coupling.”

  1. The function of Pentoxifylline binds to A2BAR in cancer seems to be of extra significance. Perhaps this paragraph can be expanded to include more information at hand.

Author responseThank you for this comment. We included a paragraph in the introduction and the discussion.

  1. (Dll4) expand on first use.

Author responseThank you for this comment. (Dll4) expanded:one of the five mammalian receptors involved in cellular regulation that also plays an integral part in sprouting angiogenesis, and its delta-like 4 (Dll4) ligand. Delta-like 4 (Dll4) is a transmembrane ligand for Notch receptors that is expressed in arterial blood vessels and sprouting endothelial cells.”

Reviewer 2 Report

The review article entitled "Parsimonious Effect of Pentoxifylline on Angiogenesis: A 2 Novel Pentoxifylline Biased Adenosine G Protein-Coupled receptor Signaling Platform" by Khoury et al. is trying to summarize the effect of known FDA-approved small molecule-Pentoxifylline which is commonly used to treat peripheral arterial disease, on angiogenesis. However, the manuscript is nicely organized and presented clearly.

It is necessary to include the data on the IC50 value of Pentoxifylline against each cell line described in the table.

Authors are also requested to perform the in-silico analysis between Pentoxifylline with A2BAR and show the reaction site that may open a future scope in drug design. 

Author Response

Reviewer #2:

Comments and Suggestions for Authors

  1. The review article entitled "Parsimonious Effect of Pentoxifylline on Angiogenesis: A 2 Novel Pentoxifylline Biased Adenosine G Protein-Coupled receptor Signaling Platform" by Khoury et al. is trying to summarize the effect of known FDA-approved small molecule-Pentoxifylline which is commonly used to treat peripheral arterial disease, on angiogenesis. However, the manuscript is nicely organized and presented clearly.

Author responseThank you for this comment.

  1. It is necessary to include the data on the IC50 value of Pentoxifylline against each cell line described in the table.

Author responseThank you for this comment. Only 3 papers made mention of the IC50 values: Kamran and Gude 2013: 7mM after 24 hrs PTX exposure to A375 human melanoma cell line; Pratibha et al 2013 which calculated 39.2 +/- 1.3 mM after PTX 2 h exposure to B16-F10 melanoma cells, and Gude et al., 2001 in C57BL/6J mice model injected with B16-F10 melanoma cells. IC50 7mM after 24 hrs PTX exposure. The other publications made no mention of IC50 values.

  1. Authors are also requested to perform the in-silico analysis between Pentoxifylline with A2BAR and show the reaction site that may open a future scope in drug design.

Author responseThank you for this important comment. We have included the in-silico analyses for PTX and A2BAR in the revised manuscript (please see lines 312-372). The dilemma is that inhibition of angiogenesis requires A2BAR antagonists, whereas inhibition of growth signaling via the ERK/MAP kinase pathway might be achieved through treatment with A2BAR agonists. It is hypothesized that the level of Gα-coupled A2A receptor expression in the naïve state influences the partitioning of its G protein and the co-expressed receptor in different membrane domains, staging the NF‐κB signaling in pentoxifylline‐mediated epigenetic effects (Figure 3).

Round 2

Reviewer 1 Report

I have no further comments.

Reviewer 2 Report

Acceptable